# OpenReview forum: "NavTrust: Benchmarking Trustworthiness for Embodied Navigation"
_ICLR.cc/2026/Conference — Submitted to ICLR 2026_

### Official Review · Reviewer_u1M3 · 2025-10-26

**Soundness:** 3
**Presentation:** 3
**Contribution:** 3
**Rating:** 6
**Confidence:** 2

**Summary:**

The paper introduces **NavTrust**, a unified benchmark for evaluating **trustworthiness and robustness** in embodied navigation across **Vision-Language Navigation (VLN)** and **Object-Goal Navigation (OGN)**.
 NavTrust integrates controlled **RGB, depth, and instruction corruptions**, evaluates six SOTA models, and compares four mitigation strategies: data augmentation, teacher–student distillation, adapter tuning, and safeguard LLM.
 The benchmark exposes significant robustness gaps and proposes a standardized, open evaluation framework.

**Strengths:**

This paper centers on a new and significant domain within embodied navigation tasks, which is important for understanding the robustness of current navigation models. The authors give a comprehensive benchmark for evaluating models and also propose several methods to enhance robustness, which will be helpful for future research.
- Comprehensive corruption coverage— unifies visual and linguistic robustness settings.
- Systematic experimental setup— six representative models evaluated with SR, SPL, and PRS metrics.
- Insightful comparison of robustness strategies — offers actionable findings for future research.
- Clear and professional writing — well-structured narrative and informative figures.

**Weaknesses:**

The main concern is that the paper relies on a single dataset (Matterport3D), which may limit the generalizability of its findings (e.g., the observed lack of robustness). The authors should include evaluations in additional environments to validate these conclusions and the proposed robustness-improving methods.

In addition, some of the methods for improving robustness lack sufficient detail.

**Questions:**

1. Can NavTrust extend to multilingual or outdoor environments (e.g., RXR), or extend to scenes except in Matterport3D?
2. How does PRS align with robustness metrics like mCE or robustness drop?
3. Is the teacher in distillation unimodal or multimodal?
4. It would be helpful to present the experimental results with a more readable visualization (e.g., consolidated figures or summary tables) to facilitate understanding of the impacts of different corruptions and methods.

---

> ### Author Response · Authors · 2025-11-23
> **Response to Reviewer u1M3 (Part 1)**
>
> First of all, we sincerely thank the reviewer for the insightful feedback. We are encouraged that the reviewer pointed out the strengths of our paper: 1) Comprehensive corruption coverage; 2) Systematic experimental setup; 3) Insightful comparison of robustness strategies; and 4) Clear and professional writing. We address the reviewer’s comments and questions point by point in the following.
>
> ## Response to Weakness 1 & Question 1
> We utilized Matterport3D [A] to benchmark heterogeneous agents under unified, challenging conditions (cluttered multi-room layouts and long-horizon instructions). To further enhance scale and diversity, we expanded NavTrust to include the multilingual RxR [B] dataset, increasing the instruction volume by an order of magnitude and the larger HM3D [C] dataset for OGN models. Full results and analysis are provided in Appendix A.1.
>
> Generalization & Dynamics: Our trends hold across RxR [B] and HM3D [C], ETPNav [D] remains robust due to high-quality instructions, while Uni-NaVid [K] (unseen languages) and PSL [J] (no depth) degrade significantly. We observe that path efficiency drops before success, implying agents adopt cautious trajectories under noise. Crucially, depth encoders collapse under Gaussian noise despite handling quantization well, identifying active sensor noise as a key bottleneck.
>
> ### RGB Corruption (SR / SPL)
>
> | Model | ETPNav | PSL | VLFM | WMNav | Uni-NaVid (New Model) | L3MVN | NaVid-7B |
> | :--- | :---: | :---: | :---: | :---: | :---: | :---: | :---: |
> | **Dataset** | RxR | HM3D | HM3D | HM3D | HM3D | HM3D | HM3D |
> | **PRS-SR** | 0.89 | 0.59 | 0.95 | 0.86 | 0.64 | 0.89 | 0.64 |
> | **PRS-SPL** | 0.87 | 0.54 | 0.93 | 0.84 | 0.64 | 0.88 | 0.64 |
> | **Uncorrupted** | 56 / 0.45 | 44 / 0.19 | 50 / 0.30 | 55 / 0.20 | 57 / 0.50 | 50 / 0.23 | 26 / 0.23 |
> | **Motion Blur** | 54 / 0.42 | 41 / 0.17 | 47 / 0.29 | 51 / 0.20 | 15 / 0.14 | 47 / 0.21 | 18.8 / 0.18 |
> | **Low-Lighting (w/o noise)** | 49 / 0.41 | 38 / 0.16 | 48 / 0.30 | 47 / 0.17 | 25 / 0.22 | 49 / 0.23 | 17.0 / 0.16 |
> | **Low-Lighting (w/ noise)** | 51 / 0.40 | 3 / 0.01 | 49 / 0.29 | 45 / 0.15 | 30 / 0.27 | 40 / 0.17 | 7.3 / 0.05 |
> | **Spatter** | 51 / 0.41 | 21 / 0.06 | 45 / 0.27 | 43 / 0.15 | 8 / 0.07 | 31 / 0.12 | 22 / 0.21 |
> | **Flare** | 52 / 0.40 | 33 / 0.14 | 50 / 0.30 | 50 / 0.18 | 34 / 0.30 | 47 / 0.22 | 22 / 0.21 |
> | **Defocus** | 51 / 0.41 | 41 / 0.17 | 48 / 0.29 | 52 / 0.18 | 33 / 0.29 | 47 / 0.22 | 25.5 / 0.23 |
> | **Black-out** | 41 / 0.29 | 17 / 0.05 | 44 / 0.24 | 46 / 0.14 | 9 / 0.07 | 50 / 0.22 | 4 / 0.01 |
>
> ### Depth Corruption
>
> | Depth Corruption | ETPNav | WMNav | VLFM | L3MVN |
> | :--- | :---: | :---: | :---: | :---: |
> | **Dataset** | RxR | HM3D | HM3D | HM3D |
> | **PRS-SR** | 0.87 | 0.87 | 0.62 | 0.56 |
> | **PRS-SPL** | 0.87 | 0.79 | 0.64 | 0.53 |
> | **Uncorrupted** | 56 / 0.45 | 55 / 0.20 | 50 / 0.30 | 50 / 0.23 |
> | **Gaussian Noise** | 53 / 0.42 | 49 / 0.15 | 0 / 0.00 | 2 / 0.01 |
> | **Missing Data** | 37 / 0.27 | 45 / 0.14 | 47 / 0.29 | 25 / 0.09 |
> | **Multipath** | 53 / 0.43 | 47 / 0.16 | 27 / 0.18 | 34 / 0.15 |
> | **Quantization** | 52 / 0.43 | 51 / 0.18 | 49 / 0.30 | 51 / 0.24 |
>
> Language Vulnerability: Base models proved surprisingly brittle to linguistic shifts. Realistic style changes (e.g., "Formal") caused performance to drop by nearly two-thirds, likely due to unseen vocabulary triggering hallucinations. Similarly, adversarial attacks and content masking severely degraded navigation, confirming that complex language instructions are a primary source of vulnerability alongside sensory corruption.
>
> ### Instruction Corruption Performance
>
> | Instruction Corruption | NaVid-7B | UniNavid | ETPNav |
> | :--- | :---: | :---: | :---: |
> | **Clean** | 46 / 0.41 | 57 / 0.50 | 57 / 0.46 |
> | **Friend** | 28 / 0.24 | 30 / 0.26 | 24 / 0.18 |
> | **Novice** | 33 / 0.26 | 33 / 0.28 | 31 / 0.21 |
> | **Formal** | 24 / 0.22 | 26 / 0.23 | 20 / 0.15 |
> | **Professional** | 20 / 0.20 | 21 / 0.20 | 17 / 0.14 |
> | **Capitalization** | 48 / 0.43 | 58 / 0.51 | 56 / 0.45 |
> | **Masking 50%** | 34 / 0.29 | 36 / 0.31 | 29 / 0.22 |
> | **Masking 100%** | 20 / 0.19 | 21 / 0.18 | 19 / 0.15 |
> | **Black-box** | 27 / 0.25 | 46 / 0.38 | 25 / 0.18 |
> | **White-box** | 30 / 0.27 | 28 / 0.24 | -- |
> | **PRS-SR** | 0.64 | 0.58 | 0.48 |
> | **PRS-SPL** | 0.64 | 0.58 | 0.46 |

---

> ### Author Response · Authors · 2025-11-23
> **Response to Reviewer u1M3 (Part 2)**
>
> Uni-Navid RXR Language Breakdown
>
>
>
> | Corruption      | EN-IN      | EN-US      | HI-IN      | TE-IN      | All (avg)   |
> |:----------------|:-----------|:-----------|:-----------|:-----------|:------------|
> | Clean           | 55 / 0.48  | 59 / 0.52  | 12 / 0.11  | 11 / 0.10  | 34 / 0.30   |
> | Black Out       | 13 / 0.11  | 14 / 0.11  | 7 / 0.06   | 2 / 0.02   | 9 / 0.07    |
> | Defocus Blur    | 52 / 0.45  | 60 / 0.53  | 13 / 0.12  | 5 / 0.05   | 33 / 0.29   |
> | Flare           | 57 / 0.50  | 59 / 0.51  | 12 / 0.11  | 8 / 0.08   | 34 / 0.30   |
> | Foreign Object  | 30 / 0.25  | 36 / 0.29  | 12 / 0.11  | 7 / 0.06   | 21 / 0.18   |
> | Low Lighting    | 34 / 0.31  | 46 / 0.41  | 9 / 0.08   | 9 / 0.08   | 25 / 0.22   |
> | Motion Blur     | 16 / 0.14  | 26 / 0.23  | 10 / 0.09  | 10 / 0.09  | 15 / 0.14   |
> | Noise           | 42 / 0.38  | 59 / 0.52  | 12 / 0.11  | 7 / 0.06   | 30 / 0.27   |
> | Spatter         | 10 / 0.09  | 5 / 0.05   | 9 / 0.09   | 8 / 0.08   | 8 / 0.07    |
>
>
> ## Response to Weakness 2 & Question 3
>
> We thank the reviewer for the comment. Detailed descriptions of these methods are currently provided in Section 4.4, Appendix A.3 and Appendix A.4 in the updated paper. We would be happy to elaborate further during the discussion period if there are specific details the reviewer finds missing or unclear.
>
> To address your specific question: The teacher is **multimodal**. The teacher is an instance of the full multimodal agent of ETPNav [D] trained on augmented datasets.
>
> ## Response to Question 2
>
> We thank the reviewer for the question. To further validate our findings, we have computed the Mean Corruption Error (mCE) [E] and the Average Robustness Drop (i.e., Success Rate on uncorrupted observation - average Success Rate on corrupted observation) for all models. Our analysis confirms that our proposed PRS (Performance Robustness Score) is strongly correlated with these established metrics. Specifically, we observe a monotonic inverse correlation between PRS, mCE, and Robustness Drop, demonstrating that high PRS scores reliably indicate low corruption error.
> For the mCE calculation, we selected the ETPNav [D] model because it is the only one capable of processing all three modalities (RGB, Depth, and Instructions), thereby providing a consistent reference point across all corruption types. The mCE and Robustness Drop results are summarized below (mCE / Robustness Drop [%] ):
> | Corruption Types  | ETPNav | NaVid | WMNav | L3MVN | PSL | VLFM |
> |------------|---------|----------|-----------|------------|----------|-------|
> | RGB Image Corruption | 1.00 / 8.88 | 1.53 / 14.63 | 0.29 / 2.25 | 1.20 / 11.50 | 2.55 / 23.00 | 0.08 / 0.38 |
> | Depth Image Corruption | 1.00 / 25 | N/A | 0.18 / 3.75 | 1.48 / 31.00 | N/A | 1.20 / 21.25 |
> | Instruction Corruption | 1.00 / 18.13 | 0.11 / 5.78 | N/A | N/A | N/A | N/A |
>
> These results corroborate our main findings: models with high PRS-SR (e.g., VLFM and WMNav) achieve significantly lower mCE scores (e.g., 0.08 and 0.29 on RGB) and Robustness Drop scores (e.g., 0.38 and 2.25 on RGB), confirming their superior stability. Conversely, models with lower PRS-SR exhibit high mCE values (e.g., PSL at 2.55 on RGB) and Robustness Drop values (e.g., PSL at 23.00 on RGB), highlighting their sensitivity to perturbations.
>
> ## Response to Question 4
>
> We thank the reviewer for the question. We have provided a summary table of the PRS-SR (Performance Robustness Score - Success Rate) across different corruption types to provide a holistic view of model performance:
>
> | Corruption Types  | ETPNav | NaVid | WMNav | L3MVN | PSL | VLFM |
> |------------|---------|----------|-----------|------------|----------|-------|
> | RGB Image Corruption | 0.86 | 0.63 | 0.94 | 0.79 | 0.47 | 0.99 |
> | Depth Image Corruption | 0.62 | N/A | 0.90 | 0.44 | N/A | 0.57 |
> | Instruction Corruption | 0.72 | 0.86 | N/A | N/A | N/A | N/A |
>
> Additionally, we have refined Figure 4 in the updated paper by bolding the text to enhance readability. Furthermore, we added new video demonstrations to the video in the supplementary materials to provide concrete visualizations of how different architectures degrade under specific corruption scenarios.

---

> ### Author Response · Authors · 2025-11-23
> **Response to Reviewer u1M3 (Part 3)**
>
> ## References:
>
> [A] Chang, Angel, et al. "Matterport3D: Learning from RGB-D Data in Indoor Environments." 2017 International Conference on 3D Vision (3DV), 2017.
>
> [B] Ku, Alexander, et al. "Room-Across-Room: Multilingual Vision-and-Language Navigation with Dense Spatiotemporal Grounding." Empirical Methods in Natural Language Processing (EMNLP). 2020.
>
> [C] Ramakrishnan, Santhosh Kumar, et al. "Habitat-Matterport 3D Dataset (HM3D): 1000 Large-scale 3D Environments for Embodied AI." Thirty-fifth Conference on Neural Information Processing Systems Datasets and Benchmarks Track, 2021.
>
> [D] An, Dong, et al. "Etpnav: Evolving topological planning for vision-language navigation in continuous environments." IEEE Transactions on Pattern Analysis and Machine Intelligence (2024).
>
> [E] Hendrycks, Dan, and Thomas Dietterich. "Benchmarking Neural Network Robustness to Common Corruptions and Perturbations." International Conference on Learning Representations. 2019
>
> [F] Nie, Dujun, et al. “Wmnav: Integrating vision-language models into world models for object goal navigation.” IEEE/RSJ International Conference on Intelligent Robots and Systems (IROS), 2025.
>
> [G] Yu, Bangguo, Hamidreza Kasaei, and Ming Cao. “L3mvn: Leveraging large language models for visual target navigation.” IEEE/RSJ International Conference on Intelligent Robots and Systems (IROS), 2023.
>
> [H] Yokoyama, Naoki, et al. “Vlfm: Vision-language frontier maps for zero-shot semantic navigation.” IEEE International Conference on Robotics and Automation (ICRA), 2024.
>
> [I] Zhang, Jiazhao, et al. “NaVid: Video-Based VLM Plans the Next Step for Vision-and-Language Navigation.” Robotics: Science and Systems, 2024.
>
> [J] Sun, Xinyu, et al. “Prioritized semantic learning for zero-shot instance navigation.” European Conference on Computer Vision, 2024.
>
> [K] Zhang, Jiazhao, et al. “Uni-NaVid: A Video-Based Vision-Language-Action Model for Unifying Embodied Navigation Tasks.” Robotics: Science and Systems, 2025.

---

> > ### Comment · Reviewer_u1M3 · 2025-11-26
> >
> > Thanks. The authors have addressed my concerns, and I will maintain my rating.

---

### Official Review · Reviewer_zNhG · 2025-10-29

**Soundness:** 3
**Presentation:** 3
**Contribution:** 2
**Rating:** 4
**Confidence:** 5

**Summary:**

This paper introduces NavTrust, a benchmark designed to evaluate the trustworthiness of embodied navigation agents, including OGN and VLN, under various corrupted and uncertain conditions. The authors propose a set of synthetic corruption settings affecting both visual and language inputs, aiming to simulate realistic challenges such as sensor noise and instruction ambiguity. Alongside the benchmark, the paper also presents mitigation strategies to improve model robustness and trust alignment. Experiments are conducted on several baseline methods to assess their sensitivity to these perturbations and the effectiveness of the proposed approaches.

**Strengths:**

1. Diverse corruption settings. The benchmark covers a wide range of corruption types—both visual (e.g., RGB and depth distortions) and linguistic (e.g., instruction noise)—providing a comprehensive way to test model robustness across different failure modes.
2. Novelty in introducing corruption-based evaluation for trust. The idea of explicitly modeling and quantifying trust under corrupted conditions for embodied navigation is novel and meaningful, which is a direction that has not been well explored in prior work.
3. Combined benchmark and mitigation strategies. The paper goes beyond benchmark design by also proposing concrete mitigation methods and demonstrating their impact. This dual contribution—offering both a testing framework and practical solutions—adds solid value to the community.

**Weaknesses:**

1. Limited connection to real-world evaluation. The benchmark is built entirely on synthetically distorted data for both training and testing. While this helps simulate ID and OOD conditions, it doesn’t clearly show how well these settings reflect real-world navigation challenges. To convincingly demonstrate the benchmark’s usefulness, I’d like to see two things: first, whether model performance on these distortions aligns with what happens in real-world cases; and second, whether the proposed mitigation strategies actually help in those real situations. Without that, the synthetic distortions feel somewhat artificial and hard to justify.
2. Baselines are too limited. The benchmark only tests two VLN methods, leaving out many others—especially the newer LLM- or VLM-based approaches that are training-free and could behave differently under such variations. For a benchmark paper, the range of baselines is too narrow, which makes it hard to judge how broadly useful or general the benchmark really is.
3. Some figures and explanations are confusing. A few parts of the presentation could be clearer. For example, Figure 1 seems to imply that RGB and depth corruptions apply only to OGN, while instruction corruption is used only for VLN, which isn’t accurate. The depth image in Figure 2 also looks odd and doesn’t resemble a typical depth map. In Figure 4, the heatmap doesn’t show the uncorrupted performance for comparison, and it’s confusing that the VLFM achieves 99% PRS yet still suffers a noticeable drop in performance. These inconsistencies make it harder to interpret the results.

**Questions:**

1. Why is SR reported without a percentage sign while SPL uses one?
2. Why does VLFM show the smallest performance drop among all baselines?
3. Since this is a benchmark paper, it would help to include some descriptive analysis—such as the number of cases per corruption category, distribution of difficulty, or other relevant statistics—to give readers a clearer understanding of the dataset’s composition and diversity.

---

> ### Author Response · Authors · 2025-11-23
> **Response to Reviewer zNhG (Part 1)**
>
> First of all, we sincerely thank the reviewer for the insightful feedback. We are encouraged that the reviewer pointed out the strengths of our paper: 1) Diverse corruption settings; 2) Novelty in introducing corruption-based evaluation for trust; and 3) Combined benchmark and mitigation strategies. We address the reviewer’s comments and questions point by point in the following. To answer your specific questions, the model performance on these distortions truly aligns with what happens in real-world cases, and the proposed mitigation strategies actually help in those real situations.
>
> ## Response to Weakness 1
>
> We thank the reviewer for the comment. We deployed ETPNav [A] and a new model (refer to Response to Weakness 2), Uni-Navid [B], on a real robot under corrupted and mitigated settings. The results and demonstration videos are now included in the supplementary material. The video includes how the model behaves in clean, corrupted, and mitigated environments.
>
> ## Response to Weakness 2
>
> We thank the reviewer for the comment. We have expanded our model suite by incorporating Uni-NaVid [B], a state-of-the-art, video-based Vision-Language-Action (VLA) model. This addition diversifies our evaluation.
> Unlike traditional modular or map-building VLN agents (i.e., building an intermediate map to aid decision-making, e.g., ETPNav [A], VLFM [C]), Uni-NaVid is a VLA that directly maps online RGB video streams and language instructions to low-level actions without intermediate map construction. This allows us to test the robustness of map-implicit methods against the map-explicit baselines. Furthermore, by including a model trained on a massive, diverse dataset (3.6M samples across 4 tasks), we can now verify if large-scale pre-training confers inherent robustness ("safety by scale") compared to specialized agents. We have reported the RGB and instruction corruption results and along with the mitigation in the instruction corruption.
> Note that Uni-NaVid [B] degrades RxR significantly compared to their paper, especially for Hindi and Telugu instructions, which are not covered in its original training data. (Appendix A.1)
>
> Uni-Navid RXR Language Breakdown  (SR/SPL)
>
>
> | Corruption      | EN-IN      | EN-US      | HI-IN      | TE-IN      | All (avg)   |
> |:----------------|:-----------|:-----------|:-----------|:-----------|:------------|
> | Clean           | 55 / 0.48  | 59 / 0.52  | 12 / 0.11  | 11 / 0.10  | 34 / 0.30   |
> | Black Out       | 13 / 0.11  | 14 / 0.11  | 7 / 0.06   | 2 / 0.02   | 9 / 0.07    |
> | Defocus Blur    | 52 / 0.45  | 60 / 0.53  | 13 / 0.12  | 5 / 0.05   | 33 / 0.29   |
> | Flare           | 57 / 0.50  | 59 / 0.51  | 12 / 0.11  | 8 / 0.08   | 34 / 0.30   |
> | Foreign Object  | 30 / 0.25  | 36 / 0.29  | 12 / 0.11  | 7 / 0.06   | 21 / 0.18   |
> | Low Lighting    | 34 / 0.31  | 46 / 0.41  | 9 / 0.08   | 9 / 0.08   | 25 / 0.22   |
> | Motion Blur     | 16 / 0.14  | 26 / 0.23  | 10 / 0.09  | 10 / 0.09  | 15 / 0.14   |
> | Noise           | 42 / 0.38  | 59 / 0.52  | 12 / 0.11  | 7 / 0.06   | 30 / 0.27   |
> | Spatter         | 10 / 0.09  | 5 / 0.05   | 9 / 0.09   | 8 / 0.08   | 8 / 0.07    |
>
> ### Uni-NaVid Instruction Corruption Performance (SR% / SPL)
>
> PE - Prompt Engineering and FT - Finetuned
>
> Beyond sensory noise, we examined the robustness of language instructions, finding that base models are surprisingly brittle to realistic stylistic changes. When exposed to different personas (such as "Formal" or "Novice"), success rates dropped heavily, losing up to two-thirds of their clean performance, this is likely because the complex and unseen vocabulary in these longer instructions triggers hallucinations. While superficial changes like capitalization had minimal impact, removing content via masking severely degraded performance, forcing agents to fall back on environmental priors rather than following the prompt. Furthermore, adversarial attacks proved highly damaging, with targeted white-box edits successfully overriding the models' inductive biases. Ultimately, these results demonstrate that as navigation instructions become longer and more complex, the language modality itself becomes a primary source of vulnerability.
>
> | Instruction Corruption | UniNavid | UniNavid-FT | UniNavid-PE |
> | :--- | :---: | :---: | :---: |
> | **Clean** | 57 / 0.50 | 57 / 0.50 | 57 / 0.50 |
> | **Friend** | 30 / 0.26 | 38 / 0.32 | 33 / 0.29 |
> | **Novice** | 33 / 0.28 | 42 / 0.34 | 32 / 0.28 |
> | **Formal** | 26 / 0.23 | 39 / 0.33 | 28 / 0.24 |
> | **Professional** | 21 / 0.20 | 40 / 0.32 | 24 / 0.21 |
> | **Capitalization** | 58 / 0.51 | -- | -- |
> | **Masking 50%** | 36 / 0.31 | -- | -- |
> | **Masking 100%** | 21 / 0.18 | -- | -- |
> | **Black-box** | 46 / 0.38 | 53 / 0.49 | 52 / 0.48 |
> | **White-box** | 28 / 0.24 | 55 / 0.48 | 54 / 0.47 |
> | **PRS-SR** | 0.58 | 0.78 | 0.65 |
> | **PRS-SPL** | 0.58 | 0.76 | 0.66 |

---

> ### Author Response · Authors · 2025-11-23
> **Response to Reviewer zNhG (Part 2)**
>
> ## Response to Weakness 3
>
> We thank the reviewer for the comment.  We have updated Figure 1 in the updated paper, which now illustrates our corruption in a clearer way (i.e., use only two panels for VLN and OGN, respectively, and clearly illustrate the possible corruption types). Also, for Figure 2, we have changed the depth image to a typical scene of a living room. For Figure 4, we would like to clarify that the “clean” row shows uncorrupted performance. For the VLFM’s [C] analysis, please refer to our Response to Question 2 below.
>
> ## Response to Question 1
>
> Thank you for the question. To clarify, as stated in Section 4.1, Success Rate (SR) is defined as a percentage, whereas Success-weighted Path Length (SPL) is a normalized metric ranging from 0 to 1. In the paper, SR is reported as a raw number (e.g., “45”) that already represents a percentage. We have made this distinction clearer in the caption of Figure 4 to avoid confusion.
>
> ## Response to Question 2
>
> Thank you for the question. We would like to clarify that we have already highlighted several reasons in Section 4.2 in the original paper. The reasons for VLFM’s [C] robustness are elaborated on further as follows:
> Our analysis suggests that VLFM's superior robustness is driven by its **modular architecture**, which decouples navigation from semantic perception, and its **foundation model backbone (BLIP-2 [D]).**
> 1) **Architectural Decoupling:** VLFM constructs its navigation graph (occupancy map) and identifies navigable frontiers solely using depth observations, which are unaffected by RGB corruptions. The agent identifies where it can move (geometric frontiers) using depth, and uses RGB (via a VLM) only to score which frontier is most relevant. When RGB inputs are severely corrupted, the VLM’s semantic scoring may degrade. However, because the underlying frontier map remains valid (constructed via depth), the agent can still navigate valid paths rather than freezing or colliding. For PSL [E] and NaVid [F], RGB pixels are direct inputs to the control policy. Consequently, when RGB shifts out of distribution, the entire planning module collapses. For L3VMN [G], though it is modular, semantic maps are generated first and then used to generate frontier maps. So if the RGB-D image is corrupted, the entire pipeline collapses.
> 2) **Foundation Model Invariance:** For its semantic scoring, VLFM employs a frozen, pre-trained vision-language model (BLIP-2 [D]). Unlike the task-specific encoders, BLIP-2 was pre-trained on massive, diverse real-world datasets containing significant noise, lighting variations, and blur. This pre-training grants VLFM a high degree of invariance to photometric corruptions, allowing it to maintain high semantic performance even when visual quality degrades.
> We have incorporated these explanations and analyses into the updated paper.
>
> ## Response to Question 3
> Thank you for the suggestion. In the updated paper, we have added a summary table reporting, for each corruption and severity level (Appendix A1):
> (i) the number of evaluation cases/episodes,
> (ii) approximate difficulty buckets (binned by SR across the models).
> We also include the same statistics for our RGB+depth corruptions, where co-occurring degradations induce different difficulty profiles than single corruptions.
>
> Difficulty is assigned per corruption type using the relative SR retention
> rc = avg SR(c)/avg SR(clean)
>
> aggregated across models:
> Easy (rc≥0.9)
> Medium (0.7≤rc<0.9)
> Hard (rc<0.7)
> |Corruption|# types|Episodes/type|Total episodes|#(Easy/Medium/Hard)|
> |-|-|-|-|-|
> |RGB|8|1000|8000|2/5/1|
> |Depth|4|1000|4000|0/1/3|
> |Instruction|9|1000|9000|1/5/3|
> |RGB+Depth|2|1000|2000|0/0/2|
>
> ## References
>
> [A] An, Dong, et al. "Etpnav: Evolving topological planning for vision-language navigation in continuous environments." IEEE Transactions on Pattern Analysis and Machine Intelligence (2024).
>
> [B] Zhang, Jiazhao, et al. ‘Uni-NaVid: A Video-Based Vision-Language-Action Model for Unifying Embodied Navigation Tasks’. Proceedings of Robotics: Science and Systems, 2025.
>
> [C] Yokoyama, Naoki, et al. "Vlfm: Vision-language frontier maps for zero-shot semantic navigation." 2024 IEEE International Conference on Robotics and Automation (ICRA). IEEE, 2024.
>
> [D] Li, Junnan, et al. "Blip-2: Bootstrapping language-image pre-training with frozen image encoders and large language models." International conference on machine learning. PMLR, 2023.
>
> [E] Sun, Xinyu, et al. "Prioritized semantic learning for zero-shot instance navigation." European Conference on Computer Vision. Cham: Springer Nature Switzerland, 2024.
>
> [F] Zhang, Jiazhao, et al. ‘NaVid: Video-Based VLM Plans the Next Step for Vision-and-Language Navigation’. Proceedings of Robotics: Science and Systems, 2024.
>
> [G] Yu, Bangguo, Hamidreza Kasaei, and Ming Cao. "L3mvn: Leveraging large language models for visual target navigation." 2023 IEEE/RSJ International Conference on Intelligent Robots and Systems (IROS). IEEE, 2023.

---

> > ### Comment · Reviewer_zNhG · 2025-11-24
> > **Response to the Authors**
> >
> > Thank you for the efforts in responding to my comments. While most of my concerns have been addressed, the most critical issue—my first weakness—remains unresolved. I look forward to the authors’ further clarification and the additional experimental results they mentioned.

---

### Official Review · Reviewer_GQf3 · 2025-10-31

**Soundness:** 3
**Presentation:** 3
**Contribution:** 2
**Rating:** 2
**Confidence:** 4

**Summary:**

This work introduces NavTrust, the first unified benchmark designed to systematically evaluate the trustworthiness and robustness of embodied navigation agents to diverse RGB-Depth corruptions and instruction variation. Unlike prior benchmarks that focus on nominal conditions, NavTrust exposes navigation systems to RGB, depth, and language perturbations to simulate sensor failures and linguistic variations encountered in real-world deployment. Based on the findings of the vulnerabilities in current widely used models, the work compared different strategies to improve the robustness of the model.

**Strengths:**

The strengths of the paper can be summarized as:
1. The paper is easy to follow, and the presentation of the research results is clear and logically structured.
2. The work focuses on trustworthiness and robustness in embodied navigation, which is an underexplored yet critical area for real-world deployment of embodied systems.
3. The work systematically integrates RGB, depth, and language corruptions into a unified evaluation platform with public available evaluation protocols, which encourages reproducibility of the work and community adoption.
3. The work provides a systematic comparison of different corruptions, followed by different mitigation strategies (data augmentation, distillation, adapters, safeguard LLM), offering practical guidance for building robust embodied systems.

**Weaknesses:**

The weakness of the paper can be summarized as:
1. Novelty not sufficient: The paper mainly builds upon existing datasets (or said, simulators / environments) and evaluation settings (e.g., Matterport3D), and the technical innovations are relatively limited. Some parts that are overclaimed as contributions are in fact implementation details or engineering tricks rather than advances.
2. Limited analysis of corruptions: Although the paper implements a wide range of corruptions (e.g., motion blur, low-lighting, flare, spatter in rgb observations), these perturbations are mostly experience-driven rather than well motivated. The study lacks deeper theoretical or analytical justification. The authors mainly provide empirical comparisons without further insight.
3. Instruction perturbations are LLM-generated; without human verification, some attacks may not reflect realistic linguistic variation. Concretely, as described in the manuscript:"We generate four stylistic variants (i.e., friendly, novice, professional, and formal) for each R2R instruction using the LLaMA-3.1 model".
4. The PRS metric only focus on one aspect of the model, i.e., success rate. Concretely, "Quantifies robustness ... where Sa,0 is agent a’s clean-split success rate and Sa,k its success rate under corruption k in a family of K corruptions...".
5. Limited qualitative results and lack of real-world validation: The qualitative analysis is limited, concretely, only one closed-loop evaluation of one model (ETPNav) is provided in the accompanying video, which offers little insight into behavior differences among agents. Moreover, the paper entirely lacks real-world experiments or sim-to-real validation, making it difficult to assess how the benchmark findings translate to real-world robot performance.

**Questions:**

1. The corruption types seem like empirically chosen. Could you elaborate on the systematic analysis or theoretical  for selecting these perturbations? My first intuition is that visual perturbations in real robots often stem from geometric changes, rather than arbitrary pixel-level manipulations as shown in the results of motion blur. Have you considered modeling perturbations from the perspective of robot motion dynamics or geometry-aware image transformations instead of purely image-domain edits?
2. The instruction variations are generated by LLaMA-3.1. Have the authors evaluated whether these generated prompts preserve semantic intent or naturalness? Could you provide more quantitative or qualitative evidence showing that these linguistic corruptions reflect realistic human input variation or adversarial behavior?
3. Since the Performance Retention Score (PRS) only measures success-rate retention, have you considered incorporating additional dimensions such as path efficiency (e.g., SPL-weighted PRS) or failure-mode weighting? Could the authors provide more quantitative results with more metrics?
4. Could the authors provide more examples or visual analyses to illustrate the behavioral differences among models under various corruptions? Additionally, do authors have plans or preliminary attempts to test NavTrust in real-world settings to assess whether robustness trends observed in simulation generalize to real-world robots?

**Details Of Ethics Concerns:**

There is no ethics concern in the reviewer's opinion.

---

> ### Author Response · Authors · 2025-11-23
> **Response to Reviewer GQf3 (Part 1)**
>
> First of all, we sincerely thank the reviewer for the insightful feedback. We are encouraged that the reviewer pointed out the strengths of our paper: 1) The paper is easy to follow; 2) The work focuses on an underexplored yet critical area; 3) The work systematically integrates corruptions into a unified evaluation platform; and 4) The work provides a comparison of different corruptions and mitigation strategies. We address the reviewer’s comments and questions point by point in the following.
>
> ## Response to Weakness 1
>
> We thank the reviewer for the comment. While NavTrust is instantiated on standard environments such as Matterport3D [A], our contributions extend well beyond modifying existing datasets or adding engineering heuristics. NavTrust introduces several components that, to our best knowledge, have not been jointly examined in embodied navigation and enable new forms of robustness analysis. **Reviewer s9Yy highlighted our unified benchmark “fills a clear gap in embodied AI evaluation”. Reviewer zNhG also commented that our work has “novelty in introducing corruption-based evaluation for trust”.** We summarize our key novelty and contributions in the following:
>
> **1) A novel, unified trustworthiness benchmark for both VLN and OGN.** While NavTrust is built upon existing datasets, we enable different evaluation settings (i.e., a comprehensive set of corruptions in different modalities) and different targets (i.e., benchmarking the trustworthiness of embodied navigation models). NavTrust provides the first systematic and unified robustness evaluation across VLN and OGN within a single framework, using matched environments, corruption families, and metrics. This unified scope allows direct comparisons between paradigms and makes cross-paradigm insights possible. Matterport3D is used because it is supported across all evaluated models.
>
> **2) Systematic study of depth-sensor robustness in indoor navigation.** Prior robustness work in navigation has focused primarily on RGB corruptions. NavTrust introduces a new, comprehensive depth corruption suite, reflecting key failure modes of state-of-the-art map-building (i.e., agents that build an intermediate map to aid decision-making) navigation agents. Existing benchmarks have not evaluated embodied navigation agents under this breadth of depth-related degradations.
>
> **3) Observation-level visual corruptions grounded in real sensor failures.** Rather than perturbing images offline, we corrupt the actual observations used for planning and control. This, for example, includes black-out sequences that probe memory reliance, and spatially varying low-light patterns that mimic realistic illumination falloff. These corruptions reflect concrete sensor and environmental failures and are generally not captured by existing vision-oriented robustness datasets.
>
> **4) Structured and diverse instruction corruptions for trustworthiness analysis.** NavTrust introduces a structured, navigation-focused instruction corruption suite spanning stylistic/personality shifts, token-level masking and casing perturbations, and adversarial/jailbreak-style clauses.
>
> **5) A unified mitigation suite for embodied navigation.** NavTrust goes beyond diagnosis and provides the first side-by-side evaluation of four robustness enhancement strategies, which span data-level, representation-level, and instruction-level defenses, none of which have been jointly evaluated in prior work on embodied navigation.

---

> ### Author Response · Authors · 2025-11-23
> **Response to Reviewer GQf3 (Part 2)**
>
> ## Response to Weakness 2 & Question 1
>
> We thank the reviewer for the comment. NavTrust provides the foundational empirical structure necessary to evaluate and develop next-generation theoretical models in embodied AI. The core contribution of NavTrust is not just creating a suite of corruptions but designing a diagnostic protocol that validates or invalidates key theoretical assumptions about embodied agents. We follow a structured taxonomy that combines (i) standardized robustness benchmarks, (ii) sensor-physics-inspired perturbations, and (iii) language robustness patterns from NLP. We clarify each dimension below:
>
> **(1) Standardized and comparable perturbations (RGB + instructions)**
>
> To ensure comparability with prior work, a subset of our RGB corruptions (e.g., defocus blur, motion blur, spatter) is inherited from RobustNav [B] and related image robustness suites (e.g., ImageNet-C [C] style corruptions). This gives NavTrust a shared vocabulary of failure modes and allows practitioners to interpret our results in the context of existing robustness studies.
>
> For instruction corruptions, we mirror standard NLP robustness taxonomies:
> - Surface-level noise such as casing changes and masked tokens,
> - Paraphrastic/stylistic rephrasings (friendly, novice, formal, professional), and
> - Adversarial / prompt-injection that adds distracting or conflicting clauses. [D]
> These categories are commonly used in text robustness and adversarial evaluation.
>
> **(2) Sensor-physics grounding for RGB and depth.**
>
> For corruptions that are unique to NavTrust, we go beyond generic pixel edits and approximate specific failure modes of on-board sensors:
>
> - For RGB, our low-light corruption is implemented using a CMOS noise model [E] with signal-dependent shot noise and signal-independent read noise, rather than arbitrary darkening. This better reflects photon-limited imaging in real robots.
> - For depth, several corruptions are motivated by active-sensing physics. For example, our multipath corruption is derived from Time-of-Flight (ToF) literature [F], where phase interference from corner reflections causes structured range errors. Thus, while implemented in the image domain, these corruptions are grounded in known sensor behavior rather than arbitrary pixel noise.
>
> **(3) Why image-domain perturbations vs. explicit motion/geometry modeling?**
>
> We agree that robot motion dynamics and geometry-aware transformations (e.g., pose noise, wheel slip, calibration errors) are important sources of failure. In this work, however, we deliberately focus NavTrust on perceptual robustness, i.e., how the policy copes with corrupted observations rather than on actuation or dynamics. Many motion-induced failures manifest visually (e.g., high-speed motion and vibration appear as motion blur; rolling-shutter distortions appear as skewed frames). By directly modeling the visual artifact (blur, noise, flare) instead of the underlying control disturbance, we isolate the robustness of the perception-policy pipeline while keeping the benchmark simulator-agnostic and easy to reproduce.
>
> Geometric changes, such as mis-calibrated intrinsics/extrinsics, are indeed valuable future extensions; they typically live in a different layer of the stack and are often addressed via calibration before deployment. Our current scope targets environmental and sensor degradations (lighting, sensor noise, occlusions) that cannot be fully calibrated away. We have clarified this design choice in the revised paper and discussed geometry and dynamics-aware perturbations as planned future directions for NavTrust.
>
> ## Response to Weakness 3 & Question 2
>
> We thank the reviewer for the comment, and we address it from two angles.
>
> **1) LLM-based judging.** We used Gemini 2.5 Pro as an LLM judge on the same set of pairs; it assigned an average semantic-preservation score of 95/100, where 5 points were deducted due to Formal and Professional overlapping in minimal cases, further indicating that the rewritten instructions remain very close in semantic meaning to the originals.
>
> **2) Human verification.** The authors manually inspected all the original-sanitized and personality instruction pairs and judged whether the navigation goal, key landmarks, and directional relations were preserved. We did not observe any cases where the task goal changed, and only minor paraphrasing occasionally omitted non-essential wording. This supports our claim that the sanitizer keeps task-relevant semantics largely intact while altering surface form.

---

> ### Author Response · Authors · 2025-11-23
> **Response to Reviewer GQf3 (Part 3)**
>
> ## Response to Weakness 4 & Question 3
>
> We thank the reviewer for the comment. To capture the nuanced trade-offs (e.g., excessive spinning or backtracking), we have introduced an **SPL-based Performance Retention Score (PRS-SPL)** in the updated paper. While PRS-SR measures the retention of task success, PRS-SPL quantifies the retention of efficiency.
>
> Comparing these two metrics allows us to isolate behaviors where an agent remains successful but becomes highly inefficient. Our analysis indicates that PRS-SPL largely aligns with PRS-SR in terms of overall trends and model rankings. For example, VLFM [G] remains the top-performing agent across both metrics (PRS-SR: 0.99, PRS-SPL: 0.88), and the performance gaps between other models (e.g., ETPNav [H], WMNav [I], and NaVid [J]) remain consistent across both scores. This alignment suggests that PRS-SR is a reliable high-level indicator of trustworthiness, while PRS-SPL provides a stricter evaluation by quantitatively penalizing the efficiency degradation - such as longer paths - that often accompanies success under corruption.
>
> ## Response to Weakness 5 & Question 4
>
> We thank the reviewer for the comment. To address this, we have made the following updates:
>
> **Comparisons Between Different Models.** We have added more video logs for the evaluated models and incorporated these visual examples into our supplementary video, which enables clearer qualitative comparisons across models.
>
> **Real-World Experiments (Sim-to-Real).** We deployed ETPNav [H], and a new baseline - Uni-NaVid [K],  on a real robot under corrupted and mitigated settings. The results and demonstration videos are now included in the supplementary material.
>
> ## Reference
>
> [A] Chang, Angel, et al. "Matterport3D: Learning from RGB-D Data in Indoor Environments." 2017 International Conference on 3D Vision (3DV), 2017.
>
> [B] Chattopadhyay, Prithvijit, et al. "Robustnav: Towards benchmarking robustness in embodied navigation." Proceedings of the IEEE/CVF International Conference on Computer Vision. 2021.
>
> [C] J. Deng, W. Dong, R. Socher, L. -J. Li, Kai Li and Li Fei-Fei, "ImageNet: A large-scale hierarchical image database," IEEE Conference on Computer Vision and Pattern Recognition, 2009.
>
> [D] Liu, Yi, et al. "Prompt injection attack against llm-integrated applications." arXiv preprint arXiv:2306.05499 (2023).
>
> [E] Wei, Kaixuan, et al. "Physics-based noise modeling for extreme low-light photography." IEEE Transactions on Pattern Analysis and Machine Intelligence, 2021
>
> [F] Jiménez, David et al. “Modeling and correction of multipath interference in time of flight cameras.” Image Vis. Comput. 2014
>
> [G] Yokoyama, Naoki, et al. "Vlfm: Vision-language frontier maps for zero-shot semantic navigation." 2024 IEEE International Conference on Robotics and Automation (ICRA), 2024.
>
> [H] An, Dong, et al. "Etpnav: Evolving topological planning for vision-language navigation in continuous environments." IEEE Transactions on Pattern Analysis and Machine Intelligence (2024).
>
> [I] Nie, Dujun, et al. “Wmnav: Integrating vision-language models into world models for object goal navigation”, IEEE/RSJ International Conference on Intelligent Robots and Systems (IROS), 2025.
>
> [J] Zhang, Jiazhao, et al. “NaVid: Video-Based VLM Plans the Next Step for Vision-and-Language Navigation.” Robotics: Science and Systems, 2024.
>
> [K] Zhang, Jiazhao, et al. “Uni-NaVid: A Video-Based Vision-Language-Action Model for Unifying Embodied Navigation Tasks.” Proceedings of Robotics: Science and Systems, 2025

---

> > ### Comment · Reviewer_GQf3 · 2025-11-25
> >
> > I appreciate the authors’ effort in preparing the rebuttal. Several of my concerns and questions have been addressed. I look forward to seeing the results from the real-world deployment. Based on the current clarification, I will increase my score to 4.

---

### Official Review · Reviewer_s9Yy · 2025-11-01

**Soundness:** 3
**Presentation:** 3
**Contribution:** 3
**Rating:** 6
**Confidence:** 3

**Summary:**

This paper introduces NavTrust, a unified benchmark for evaluating the trustworthiness and robustness of embodied navigation systems across two major tasks: Vision-Language Navigation (VLN) and Object-Goal Navigation (OGN). The benchmark systematically injects corruptions into RGB, depth, and language instruction modalities to simulate realistic sensing and communication failures. It evaluates six state-of-the-art models and compares four robustness enhancement strategies—data augmentation, teacher-student distillation, adapter tuning, and LLM fine-tuning—under a standardized evaluation protocol. Extensive experiments reveal consistent performance degradation under perceptual and linguistic corruptions, providing insights into model weaknesses and practical guidelines for improving reliability.

**Strengths:**

Novel Benchmark and Scope: The paper fills a clear gap in embodied AI evaluation by jointly assessing perceptual and linguistic robustness under a unified framework. Prior works such as RobustNav and EmbodiedBench handle only subsets of these aspects.

Comprehensive Corruption Suite: The authors design realistic and diverse corruptions across RGB, depth, and instruction modalities, covering noise, occlusions, adversarial prompts, and stylistic rephrasings.

Empirical Breadth: Evaluation spans six recent navigation models (e.g., ETPNav, NaVid, VLFM, WMNav) and presents clear analyses of their relative robustness, revealing valuable comparative insights.

Constructive Contribution: The study goes beyond diagnosis by benchmarking four mitigation strategies (data augmentation, knowledge distillation, adapter tuning, LLM-based sanitization), which makes the work more actionable for practitioners.

Presentation Quality: The paper is generally well-written, with clear figures (e.g., Figure 2 illustrating corruption types and Figure 4 showing PRS comparisons) that effectively communicate experimental findings.

**Weaknesses:**

Limited Theoretical Depth: While empirically comprehensive, the paper lacks a theoretical analysis of why certain models fail under specific corruption types (e.g., the deeper mechanisms linking architecture and robustness).

Benchmark Generality: The benchmark is limited to the Matterport3D-based environments and English instructions (R2R dataset). This may restrict its generalizability to other datasets or real-world robotic settings.

Evaluation of Mitigation Strategies: The four robustness strategies are compared in aggregate but not deeply analyzed—for example, ablation studies isolating which aspects of each technique contribute most to PRS improvement would strengthen the work.

Clarity of Metric Interpretation: Although PRS (Performance Retention Score) is useful, it may oversimplify nuanced robustness trade-offs. Discussion on sensitivity or threshold effects could improve interpretability.

Missing Discussion on Computational Overhead: The additional training and evaluation cost of corruptions and mitigations is not discussed, which is relevant for benchmark adoption.

**Questions:**

See Weaknesses

---

> ### Author Response · Authors · 2025-11-23
> **Response to Reviewer s9Yy (Part 1)**
>
> First of all, we sincerely thank the reviewer for the insightful feedback. We are encouraged that the reviewer pointed out the strengths of our paper: 1) Novelty: The paper fills a clear gap in embodied AI evaluation; 2) Comprehensive Corruption Suite: Design of realistic and diverse corruptions; 3) Empirical Breath: Evaluation spans six recent navigation models; 4) Constructive Contribution: This study benchmarks four mitigation strategies; and 5) Presentation Quality: The paper is generally well-written with clear figures. We address the reviewer’s comments and questions point by point in the following.
>
> ## Response to Weakness 1
>
> Thank you for the helpful suggestion and for recognizing the empirical breadth of our work. We would like to clarify that our analysis does extend beyond reporting empirical results. Section 4.2 provides detailed mechanistic explanations linking architectural design choices to the observed robustness and failure behaviors. For convenience, we summarize the key analyses below.
>
> *1) Memory vs. Reactive Architectures.* We explain the divergence under Black-out corruptions through the role of spatial memory. Map-building agents (i.e., building an intermediate map to aid decision making, like ETPNav [A], L3MVN [B]) remain robust to sensor loss because their occupancy grids preserve past observations, whereas reactive, mapless agents (NaVid [C], PSL [D]) fail catastrophically due to the absence of a mechanism to recall missing frames. Conversely, we also analyze why map-building agents fail sharply under depth corruptions: their reliance on accurate range data makes occupancy-grid construction highly sensitive to depth noise.
>
> *2) VLM-based vs. Non-VLM Designs (Tokenization & Vocabulary).* Our analysis highlights a fundamental architectural divergence between large pre-trained VLM-based models (e.g., NaVid [C]) and task-specific navigation agents (e.g., ETPNav [A]). ETPNav’s vulnerability to stylistic changes and synonyms stems from its rigid, fixed-size tokenizer, while VLM-based agents benefit from broad semantic coverage and robust embedding spaces learned during large-scale pretraining.
>
> *3) Early vs. Late Fusion.* We compare sensor fusion mechanisms and show that early-fusion models (e.g., ETPNav [A]), which directly ingest depth, propagate noise into the planning state, making them vulnerable to depth corruptions. In contrast, late-fusion architectures like WMNav [E] first extract monocular features and use depth as an auxiliary channel with learned confidence gating, enabling them to down-weight unreliable sensor inputs and maintain robustness.
>
> These analyses, supported by the quantitative evidence in Section 4.2, demonstrate that our benchmark enables researchers to understand not only whether models fail but also why different architectures exhibit distinct robustness profiles.

---

> ### Author Response · Authors · 2025-11-23
> **Response to Reviewer s9Yy (Part 2)**
>
> ## Response to Weakness 2
>
> We thank the reviewer for the comment. To strengthen the benchmark’s generality, we have done the additional experiments that broaden its scope along two key dimensions.
>
> 1) Dataset Expansion (HM3D [F] & RxR [G]): To move beyond Matterport3D [H], we are evaluating OGN models on the HM3D dataset [F], which offers more diverse and photorealistic scenes. For VLN, we are incorporating the RxR (Room-Across-Room) dataset [G] to assess robustness under multilingual and more fine-grained instruction formulations. Given the extensive number of episodes and variations in evaluation throughput, we have completed the evaluation for several models and present the results here. We have included in-depth analysis with the full results in the revised paper (Appendix Section A.1).
>
> Generalization & Dynamics: Our trends hold across RxR [G] and HM3D [F], ETPNav [A] remains robust due to high-quality instructions, while Uni-NaVid [I] (unseen languages) and PSL [D] (no depth) degrade significantly. We observe that path efficiency drops before success, implying agents adopt cautious trajectories under noise. Crucially, depth encoders collapse under Gaussian noise despite handling quantization well, identifying active sensor noise as a key bottleneck.
>
> ### RGB Corruption (SR / SPL)
>
> | Model | ETPNav | PSL | VLFM | WMNav | Uni-NaVid (New Model) | L3MVN | NaVid-7B |
> | :--- | :---: | :---: | :---: | :---: | :---: | :---: | :---: |
> | **Dataset** | RxR | HM3D | HM3D | HM3D | HM3D | HM3D | HM3D |
> | **PRS-SR** | 0.89 | 0.59 | 0.95 | 0.86 | 0.64 | 0.89 | 0.64 |
> | **PRS-SPL** | 0.87 | 0.54 | 0.93 | 0.84 | 0.64 | 0.88 | 0.64 |
> | **Uncorrupted** | 56 / 0.45 | 44 / 0.19 | 50 / 0.30 | 55 / 0.20 | 57 / 0.50 | 50 / 0.23 | 26 / 0.23 |
> | **Motion Blur** | 54 / 0.42 | 41 / 0.17 | 47 / 0.29 | 51 / 0.20 | 15 / 0.14 | 47 / 0.21 | 18.8 / 0.18 |
> | **Low-Lighting (w/o noise)** | 49 / 0.41 | 38 / 0.16 | 48 / 0.30 | 47 / 0.17 | 25 / 0.22 | 49 / 0.23 | 17.0 / 0.16 |
> | **Low-Lighting (w/ noise)** | 51 / 0.40 | 3 / 0.01 | 49 / 0.29 | 45 / 0.15 | 30 / 0.27 | 40 / 0.17 | 7.3 / 0.05 |
> | **Spatter** | 51 / 0.41 | 21 / 0.06 | 45 / 0.27 | 43 / 0.15 | 8 / 0.07 | 31 / 0.12 | 22 / 0.21 |
> | **Flare** | 52 / 0.40 | 33 / 0.14 | 50 / 0.30 | 50 / 0.18 | 34 / 0.30 | 47 / 0.22 | 22 / 0.21 |
> | **Defocus** | 51 / 0.41 | 41 / 0.17 | 48 / 0.29 | 52 / 0.18 | 33 / 0.29 | 47 / 0.22 | 25.5 / 0.23 |
> | **Black-out** | 41 / 0.29 | 17 / 0.05 | 44 / 0.24 | 46 / 0.14 | 9 / 0.07 | 50 / 0.22 | 4 / 0.01 |
>
> ### Depth Corruption
>
> | Depth Corruption | ETPNav | WMNav | VLFM | L3MVN |
> | :--- | :---: | :---: | :---: | :---: |
> | **Dataset** | RxR | HM3D | HM3D | HM3D |
> | **PRS-SR** | 0.87 | 0.87 | 0.62 | 0.56 |
> | **PRS-SPL** | 0.87 | 0.79 | 0.64 | 0.53 |
> | **Uncorrupted** | 56 / 0.45 | 55 / 0.20 | 50 / 0.30 | 50 / 0.23 |
> | **Gaussian Noise** | 53 / 0.42 | 49 / 0.15 | 0 / 0.00 | 2 / 0.01 |
> | **Missing Data** | 37 / 0.27 | 45 / 0.14 | 47 / 0.29 | 25 / 0.09 |
> | **Multipath** | 53 / 0.43 | 47 / 0.16 | 27 / 0.18 | 34 / 0.15 |
> | **Quantization** | 52 / 0.43 | 51 / 0.18 | 49 / 0.30 | 51 / 0.24 |
>
> Language Vulnerability: Base models proved surprisingly brittle to linguistic shifts. Realistic style changes (e.g., "Formal") caused performance to drop by nearly two-thirds, likely due to unseen vocabulary triggering hallucinations. Similarly, adversarial attacks and content masking severely degraded navigation, confirming that complex language instructions are a primary source of vulnerability alongside sensory corruption.
>
> ### Instruction Corruption Performance
>
> | Instruction Corruption | NaVid-7B | UniNavid | ETPNav |
> | :--- | :---: | :---: | :---: |
> | **Clean** | 46 / 0.41 | 57 / 0.50 | 57 / 0.46 |
> | **Friend** | 28 / 0.24 | 30 / 0.26 | 24 / 0.18 |
> | **Novice** | 33 / 0.26 | 33 / 0.28 | 31 / 0.21 |
> | **Formal** | 24 / 0.22 | 26 / 0.23 | 20 / 0.15 |
> | **Professional** | 20 / 0.20 | 21 / 0.20 | 17 / 0.14 |
> | **Capitalization** | 48 / 0.43 | 58 / 0.51 | 56 / 0.45 |
> | **Masking 50%** | 34 / 0.29 | 36 / 0.31 | 29 / 0.22 |
> | **Masking 100%** | 20 / 0.19 | 21 / 0.18 | 19 / 0.15 |
> | **Black-box** | 27 / 0.25 | 46 / 0.38 | 25 / 0.18 |
> | **White-box** | 30 / 0.27 | 28 / 0.24 | -- |
> | **PRS-SR** | 0.64 | 0.58 | 0.48 |
> | **PRS-SPL** | 0.64 | 0.58 | 0.46 |
>
>
> Uni-NaVid Unseen Language Breakdown
>
> | Corruption | HI-IN | TE-IN |
> | :--- | :--- | :--- |
> | Clean | 12 / 0.11 | 11 / 0.10 |
> | Black Out | 7 / 0.06 | 2 / 0.02 |
> | Defocus Blur | 13 / 0.12 | 5 / 0.05 |
> | Flare | 12 / 0.11 | 8 / 0.08 |
> | Foreign Object | 12 / 0.11 | 7 / 0.06 |
> | Low Lighting | 9 / 0.08 | 9 / 0.08 |
> | Motion Blur | 10 / 0.09 | 10 / 0.09 |
> | Noise | 12 / 0.11 | 7 / 0.06 |
> | Spatter | 9 / 0.09 | 8 / 0.08 |
>
> 2) Real-World Experiments (Sim-to-Real):  We deployed ETPNav [A] and a new baseline - UniNaVid [I] on a real robot under corrupted and mitigated settings. The results and demonstration videos are now included in the supplementary material.

---

> ### Author Response · Authors · 2025-11-23
> **Response to Reviewer s9Yy (Part 3)**
>
> ## Response to Weakness 3
>
> Thank you for the helpful suggestion. We would like to clarify that Section 4.4 already includes several ablation-style analyses designed to isolate the specific components contributing to PRS improvements.
>
> *1) Temporal structure of augmentation.* We directly compared per-frame vs. per-episode augmentation (see Table 2) to isolate the role of temporal coherence. The results show that maintaining consistent corruptions across an episode is crucial (PRS 0.92 vs. 0.89), as breaking temporal consistency disrupts the topological graph update process.
>
> *2) Augmentation distribution and intensity.* We further evaluated the Success Rate distributed augmentation strategy (i.e., oversamples underperforming corruptions) and varied training intensities. These experiments isolate the effect of corruption scheduling and exposure frequency on the robustness gains.
>
> *3) Depth-encoder isolation.* The depth-adapter experiment explicitly targets only the depth-encoding layers, demonstrating that the performance improvement originates from correcting depth-specific brittleness rather than from general model tuning.
>
> Together, these analyses serve as ablation studies that disentangle the contributions of different components within our mitigation strategies. To further address your comment, we also added the detailed statistics on how the mitigated ETPNav [A] performs under each corruption below.
>
> (DA: Data Augmentation, PF: Per-Frame, PE: Per-Episode)
>
> | Corruption | Adapters | DA PF (σ = 0.6) | DA PE | DA SR | DA PE (σ = 0.9) | T-S distillation |
> |:-|:-:|:-:|:-:|:-:|:-:|:-:|
> | **PRS-SR (Image)**| 0.33    | 0.89| 0.92           | 0.93              | 0.94    | 0.93       |
> | **Clean (Image/Depth)**   | 65  | 65 | 65 | 65 | 65      | 65         |
> | Motion blur (Image) | 16      | 52| 66 | 60| 66      | 62         |
> | Low-light (w/o noise)| 22      | 62| 62 | 59 | 62      | 61         |
> | Low-light (w/ noise)  | 30      | 58 | 55| 64| 60      | 55         |
> | Spatter (Image)| 16      | 59   | 62| 58| 55      | 66         |
> | Flare (Image)| 24      | 62   | 60 | 64| 63      | 56         |
> | Defocus (Image)| 14      | 51| 60 | 61| 62      | 59         |
> | Foreign object (Image) | 21| 59 | 60| 59| 62      | 61         |
> | Black-out (Image)  | 26      | 58  | 52| 59| 57      | 61         |
> | **PRS (Depth)** | 0.89    | 0.67 | 0.72| 0.73              | 0.75    | 0.85       |
> | Gaussian noise (Depth)    | 55 | 33           | 59             | 38                | 42      | 42         |
> | Missing data (Depth) | 54      | 51           | 25             | 32                | 29      | 66         |
> | Multipath (Depth)| 62| 31           | 43             | 56                | 62      | 61         |
> | Quantization (Depth)| 60      | 59           | 61             | 63                | 63      | 52         |
>
> | Instruction Variant | LLaMA fine-tuning | o3 prompt engineering |
> |-|-|-|
> | **PRS** | **0.84**| **0.80**|
> | Friend| 54 | 49|
> | Novice| 52| 47|
> | Formal| 44| 43|
> | Professional| 53| 49|
> | Black Box| 63| 63|
> | White Box| 62| 61|
>
> The detailed breakdown (Appendix A.3) reveals distinct optimal strategies for different sensor modalities. Adapters emerge as the superior strategy for Depth consistency (PRS-Depth 0.89). Notably, the Depth Adapter demonstrates a robust all-around performance with no significant weaknesses, achieving consistently effective mitigation across all corruption types, whereas other methods suffer marked performance drops in specific scenarios. T-S Distillation provides the most balanced unified solution across depth and RGB mitigation; it achieves near-state-of-the-art performance on Image benchmarks (PRS 0.93) while maintaining strong Depth scores (PRS 0.85), suggesting that distillation preserves the semantic priors necessary to handle incomplete sensor inputs.
>
> When we compare these mitigated results for ETPNav to the original non-mitigated scores in Figure 4 , we see that the gains are not only statistically significant in terms of PRS but also operationally meaningful. In the original setting, ETPNav achieves a PRS-SR (Image) of 0.86 and a PRS (Depth) of 0.62. With our mitigation strategies, PRS-SR (Image) increases to 0.93–0.94 (for DA per-episode and T-S distillation), while PRS (Depth) rises as high as 0.89 with depth adapters (and 0.85 with T-S distillation), corresponding to roughly +9% relative improvement in RGB robustness and +45% in depth robustness. Importantly, these gains are driven by large improvements under the hardest corruptions rather than only marginal gains on the easy cases. For example, for RGB motion blur and low-light with noise, ETPNav’s SR improves from 57→66 and 48→60 respectively, and the worst-case image corruption SR increases from 48 to 55. On the depth side, the adapter-based mitigation raises SR under Gaussian noise from 33→55 and under missing data from 24→54, effectively eliminating the catastrophic failures previously observed for these conditions.

---

> ### Author Response · Authors · 2025-11-23
> **Response to Reviewer s9Yy (Part 4)**
>
> ## Response to Weakness 4
>
> We thank the reviewer for the insightful suggestion. To provide the sensitivity analysis and better capture nuanced robustness trade-offs, we are incorporating two additional experiments:
>
> *1) Multi-level corruption intensities.* To reveal sensitivity curves and potential failure thresholds, we are expanding our evaluation from a single intensity level to a spectrum of intensities (0.0, 0.25, 0.5, 1.0) for all corruption types. This enables a clearer interpretation of how different architectures degrade as corruption severity increases.  We show PRS-SR and PRS-SPL results below.
>
> RGB Robustness Tiers Expanded intensity analysis reveals distinct robustness tiers. VLFM shows exceptional resilience (maintaining 85% stability at 0.75 intensity), suggesting foundation models provide a semantic buffer that PSL lacks. ETPNav[A] and L3MVN degrade smoothly, whereas WMNav[E] deteriorates rapidly under dense artifacts. The instruction-heavy NaVid[C] and UniNavid[I] prove the most brittle, decaying sharply by 0.75 intensity, exposing a reliance on high-fidelity appearance. Intriguingly, mild corruptions occasionally improved path efficiency, acting as implicit regularizers against high-frequency clutter.
>
> Depth Modality Dichotomy Depth corruptions uncover a sharp contrast in geometric reliance. For ETPNav[A] and WMNav[E], the depth channel acts as a resilient backbone (>80% stability) even when RGB fails. In contrast, L3MVN[B] and VLFM[J] collapse under stochastic Gaussian noise yet tolerate structured artifacts like quantization. This indicates their encoders are calibrated for piecewise-smooth geometry, making them robust to low resolution but critically vulnerable to pixel-wise sensor noise.
>
> We have in-depth multi-intensity per corruption level in the revised research paper (Appendix A.2) with deeper analysis and figure.
>
> ### Avg PRS-SR
>
> | Model    | Modality | 0.25 | 0.50 | 0.75 | 1.00 |
> |----------|----------|------|------|------|------|
> | ETPNav   | RGB      | 0.95 | 0.89 | 0.87 | 0.60 |
> | ETPNav   | Depth    | 0.94 | 0.87 | 0.85 | 0.81 |
> | PSL      | RGB      | 0.84 | 0.59 | 0.31 | 0.04 |
> | VLFM     | RGB      | 0.95 | 0.95 | 0.85 | 0.31 |
> | VLFM     | Depth    | 0.66 | 0.62 | 0.55 | 0.51 |
> | WMNav    | RGB      | 0.94 | 0.86 | 0.62 | 0.30 |
> | WMNav    | Depth    | 0.94 | 0.87 | 0.71 | 0.61 |
> | L3MVN    | RGB      | 0.95 | 0.89 | 0.74 | 0.48 |
> | L3MVN    | Depth    | 0.71 | 0.56 | 0.44 | 0.36 |
> | UniNavid | RGB      | 0.67 | 0.64 | 0.46 | 0.23 |
> | NaVid-7B | RGB      | 0.78 | 0.64 | 0.44 | 0.21 |
>
> ### Avg PRS-SPL
>
> | Model    | Modality | 0.25 | 0.50 | 0.75 | 1.00 |
> |----------|----------|------|------|------|------|
> | ETPNav   | RGB      | 0.97 | 0.87 | 0.83 | 0.47 |
> | ETPNav   | Depth    | 0.95 | 0.87 | 0.82 | 0.78 |
> | PSL      | RGB      | 0.79 | 0.54 | 0.27 | 0.02 |
> | VLFM     | RGB      | 0.96 | 0.93 | 0.77 | 0.27 |
> | VLFM     | Depth    | 0.69 | 0.64 | 0.58 | 0.53 |
> | WMNav    | RGB      | 0.94 | 0.84 | 0.61 | 0.34 |
> | WMNav    | Depth    | 0.85 | 0.79 | 0.63 | 0.56 |
> | L3MVN    | RGB      | 0.96 | 0.88 | 0.74 | 0.33 |
> | L3MVN    | Depth    | 0.71 | 0.53 | 0.39 | 0.33 |
> | UniNavid | RGB      | 0.67 | 0.64 | 0.47 | 0.24 |
> | NaVid-7B | RGB      | 0.71 | 0.64 | 0.48 | 0.21 |
>
> *2) SPL-based Performance Retention Score (PRS-SPL).* To capture efficiency-related behaviors (e.g., unnecessary spinning, backtracking) that SR alone cannot reflect, we introduced a PRS computed using SPL. While PRS-SR quantifies retention of success, PRS-SPL measures retention of efficiency.
>
> Comparing these two metrics allows us to isolate behaviors where an agent remains successful but becomes highly inefficient. Our analysis indicates that PRS-SPL largely aligns with PRS-SR in terms of overall trends and model rankings. For example, VLFM [J] remains the top-performing agent across both metrics (PRS-SR: 0.99, PRS-SPL: 0.88), and the performance gaps between other models (e.g., ETPNav [A], WMNav [E], and NaVid [C]) remain consistent across both scores. This alignment suggests that PRS-SR is a reliable high-level indicator of trustworthiness, while PRS-SPL provides a stricter evaluation by quantitatively penalizing the efficiency degradation, such as longer paths, that often accompanies success under corruption.

---

> ### Author Response · Authors · 2025-11-23
> **Response to Reviewer s9Yy (Part 5)**
>
> ## Response to Weakness 5
>
> We thank the reviewer for the comment. In Section 4.3 of the paper, we have added a “Computational Cost and Overhead” section that reports the mean time of one episode in our NavTrust suite compared to a standard “clean” evaluation.
>
> Our profiling indicates that visual corruptions introduce significant computational overhead, while ETPNav [A] remains the most efficient architecture in absolute terms, it exhibits high relative sensitivity to noise-heavy corruptions (e.g., Low-light w/ noise increases latency by  around 5.5 times). We observe a similar trend in L3MVN, where high-frequency noise causes an extreme latency spike (up to 309s). In contrast, WMNav demonstrates remarkable temporal stability across all corruptions (consistent around 40-44s), suggesting its topological memory approach is computationally resilient to visual artifacts even when the underlying sensory data is degraded.
>
> | Corruption| ETPNav |PSL | NaVid-7B | UniNaVid | WMNav | L3MVN |  VLFM |
> |-|-:|-:|-:|-:|-:|------:|------:|
> | Clean                    |  1.01  |  10.2 |    223.0 |     15.0 |  43.8 |  38.5 |  76.7 |
> | Motion blur              |  2.53  |  15.2 |    248.0 |     19.0 |  41.5 |   2.5 |  79.8 |
> | Low-light (w/o noise)    |  2.15  |  12.6 |    163.0 |     17.0 |  40.3 |  38.2 |  75.6 |
> | Low-light (w/ noise)     |  5.59  |  73.8 |    415.0 |     29.3 |  41.5 | 309.1 |  89.4 |
> | Spatter                  |  3.28  |  31.7 |    330.0 |     33.8 |  41.5 |  28.8 |  84.7 |
> | Flare                    |  3.39  |  56.1 |    347.0 |     36.3 |  42.6 |  28.6 |  92.1 |
> | Defocus                  |  3.99  |  30.6 |    244.0 |     27.5 |  41.5 |  32.2 |  83.6 |
> | Foreign object           |  2.37  |  14.9 |    355.0 |     23.9 |  40.3 |  29.2 |  79.3 |
> | Black-out                |  2.72  |  16.5 |    378.0 |     25.9 |  40.3 |  58.6 |  84.9 |
>
>
> References
>
> [A] An, Dong, et al. "Etpnav: Evolving topological planning for vision-language navigation in continuous environments." IEEE Transactions on Pattern Analysis and Machine Intelligence, 2024.
>
> [B] Yu, Bangguo, Hamidreza Kasaei, and Ming Cao. "L3mvn: Leveraging large language models for visual target navigation." IEEE/RSJ International Conference on Intelligent Robots and Systems (IROS), 2023.
>
> [C] Zhang, Jiazhao, et al. ‘NaVid: Video-Based VLM Plans the Next Step for Vision-and-Language Navigation’. Proceedings of Robotics: Science and Systems, 2024.
>
> [D] Sun, Xinyu, et al. "Prioritized semantic learning for zero-shot instance navigation." European Conference on Computer Vision, 2024.
>
> [E] Nie, Dujun, et al. “Wmnav: Integrating vision-language models into world models for object goal navigation”, IEEE/RSJ International Conference on Intelligent Robots and Systems (IROS), 2025.
>
> [F] Ramakrishnan, Santhosh Kumar, et al. "Habitat-Matterport 3D Dataset (HM3D): 1000 Large-scale 3D Environments for Embodied AI." Thirty-fifth Conference on Neural Information Processing Systems Datasets and Benchmarks Track, 2021.
>
> [G] Ku, Alexander, et al. "Room-Across-Room: Multilingual Vision-and-Language Navigation with Dense Spatiotemporal Grounding." Empirical Methods in Natural Language Processing (EMNLP). 2020.
>
> [H] Chang, Angel, et al. "Matterport3D: Learning from RGB-D Data in Indoor Environments." 2017 International Conference on 3D Vision (3DV), 2017.
>
> [I] Zhang, Jiazhao, et al. ‘Uni-NaVid: A Video-Based Vision-Language-Action Model for Unifying Embodied Navigation Tasks’. Robotics: Science and Systems, 2025.
>
> [J] Yokoyama, Naoki, et al. "Vlfm: Vision-language frontier maps for zero-shot semantic navigation." 2024 IEEE International Conference on Robotics and Automation (ICRA), 2024.
>
> [K] Sun, Xinyu, et al. “Prioritized semantic learning for zero-shot instance navigation.” European Conference on Computer Vision, 2024.

---

### Official Review · Reviewer_mYxG · 2025-11-03

**Soundness:** 3
**Presentation:** 3
**Contribution:** 2
**Rating:** 6
**Confidence:** 4

**Summary:**

The paper introduces NavTrust, a benchmark designed to systematically evaluate the trustworthiness and robustness of embodied navigation systems. It provides a unified framework for assessing how RGB, depth, and language modalities influence navigation performance under various types of input corruption. The benchmark covers both Vision-Language Navigation and Object-Goal Navigation settings, applying controlled perturbations such as visual noise, depth sensor degradation, and instruction modification. Six state-of-the-art navigation models are benchmarked using metrics such as Success Rate (SR), Success weighted by Path Length (SPL), and a newly proposed Performance Retention Score (PRS), while four robustness enhancement strategies—data augmentation, teacher–student distillation, adapter tuning, and instruction-level defenses—are compared.

**Strengths:**

1. The paper establishes a unified and open-source benchmark that evaluates both VLN and OGN tasks. It jointly considers RGB, depth, and language modalities, offering a comprehensive view of multimodal robustness in embodied navigation.
	2. Each modality includes a wide range of realistic corruptions—visual noise, depth sensor degradation, and linguistic perturbations—that better reflect the challenges encountered in real-world navigation scenarios.
	3. Provides several mainstream robustness enhancement strategies and conducts corresponding experiments to evaluate their effectiveness.

**Weaknesses:**

1. The evaluation of depth corruptions focuses mainly on mapping-based methods. Including a broader range of approaches would provide a more comprehensive understanding of depth robustness.

2. In the mitigation stage, the analysis is conducted only on ETPNav, while comparisons across more models would strengthen the conclusions.

**Questions:**

1. In Table 1, WMNav appears to include depth sensing, but it is not shown as such in Figure 1.

2. In Figure 4 (Image Corruption), the performance of NaVid under motion blur degradation drops more than that of ETPNav, which seems inconsistent with the description around line 341.

3. The statement around line 403, claiming that “tokenization artifacts (masking, capitalization) and vocabulary coverage dominate the robustness, more so than downstream spatial reasoning failures,” seems not directly supported by experiments. The paper does not include explicit comparisons or ablations related to spatial reasoning, so this contrast may be overstated or require additional evidence.

---

> ### Author Response · Authors · 2025-11-23
> **Response to Reviewer mYxG (Part 1)**
>
> First of all, we sincerely thank the reviewer for the insightful feedback. We are encouraged that the reviewer pointed out the strengths of our paper: 1) our paper offers a comprehensive view of multimodal robustness; 2) each modality includes a wide range of corruptions; and 3) our work provides several robustness enhancement strategies. We address the reviewer’s comments and questions point by point in the following.
>
> ## Response to Weakness 1
>
> We thank the reviewer for the comment. We acknowledge that our original wording may have introduced ambiguity, and we have clarified the terminology in the revised paper by **revising “mapping-based” and “map-centric” to “map-building” (i.e., building an intermediate map representation to aid decision making)** methods. To clarify, none of the six approaches evaluated in our study relies on a pre-built or externally provided map.
>
> While we acknowledge that some end-to-end baselines ingest depth merely as a latent feature channel [A] [B], they generally do not achieve competitive performance compared to methods in our benchmark. Furthermore, although we identified a recent work [C] that effectively utilizes depth without explicit map-building for the whole environment, its implementation is not open-sourced, which prevented us from including it as a reproducible baseline. We have incorporated these references into the related work section of the revised paper.
>
> Given that depth signals are primarily used for local environment perception, online map construction, and traversability reasoning, we believe our current depth corruption evaluation provides a representative and meaningful assessment of depth robustness in the dominant usage scenarios. However, if the reviewer could point to specific state-of-the-art, open-source navigation models that use depth without building intermediate map representations, we would be more than happy to incorporate them into our comparison.
>
> ## Response to Weakness 2
>
> We thank the reviewer for the constructive suggestion. As discussed in Section 4, except for ETPNav [D], all of our evaluated models either are **zero-shot foundation-model agents that do not require training** (WMNav [E], L3MVN [F], VLFM [G]) or **do not provide publicly available training code or infrastructure** (NaVid [H], PSL [I]).
>
> To further demonstrate the generalizability of our approach, we have integrated evaluating instruction mitigation strategies to the multilingual RxR dataset, along with the recent addition of Uni-NaVid[J]. Refer to Appendix A.1 for the full result and analysis.
>
> Instruction mitigation strategies significantly improve robustness against instruction corruptions for all three models, with fine-tuning proving particularly effective. This approach yielded the largest gains, raising robustness scores by approximately 20-30% across the board and nearly doubling performance for ETPNav. Even the lighter-weight prompt engineering method delivered consistent stability improvements. Crucially, these gains extend beyond handling simple rephrasing; the models demonstrated genuine hardening against aggressive adversarial attacks, recovering 15–30 absolute percentage points in success rate. This confirms that the proposed mitigations effectively protect agents against both natural linguistic variations and severe instructional perturbations.
>
> Combined Instruction Corruption Performance (SR / SPL, FT: fine-tuned, PE: prompt-engineered):
>
> |Corruption|NaVid-7B|NaVid-7B-FT|NaVid-7B-PE|UniNavid|UniNavid-FT|UniNavid-PE|ETPNav|ETPNav-FT|ETPNav-PE|
> |-|-|-|-|-|-|-|-|-|-|
> |Clean|46 / 0.41|46 / 0.41|46 / 0.41|57 / 0.50|57 / 0.50|57 / 0.50|57 / 0.46|57 / 0.46|57 / 0.46|
> |Friend|28 / 0.24|31 / 0.24|27 / 0.21|30 / 0.26|38 / 0.32|33 / 0.29|24 / 0.18|40 / 0.20|40 / 0.30|
> |Novice|33 / 0.26|35 / 0.25|26 / 0.20|33 / 0.28|42 / 0.34|32 / 0.28|31 / 0.21|52 / 0.42|38 / 0.30|
> |Formal|24 / 0.22|30 / 0.24|23 / 0.18|26 / 0.23|39 / 0.33|28 / 0.24|20 / 0.15|39 / 0.37|32 / 0.26|
> |Professional|20 / 0.20|31 / 0.24|21 / 0.17|21 / 0.20|40 / 0.32|24 / 0.21|17 / 0.14|42 / 0.34|30 / 0.24|
> |Capitalization|48 / 0.43|-|-| 58 / 0.51|-|-|56 / 0.45|-|-|
> |Masking 50%|34 / 0.29|-|-|36 / 0.31|-|-|29 / 0.22|-|-|
> |Masking 100%|20 / 0.19|-|-|21 / 0.18 |- |- | 19 / 0.15 |- |- |
> |Black-box| 27 / 0.25|44 / 0.42 | 43 / 0.41 | 46 / 0.38 | 53 / 0.49 | 52 / 0.48 | 25 / 0.18 | 55 / 0.41 | 54 / 0.40 |
> |White-box| 30 / 0.27|45 / 0.41 | 44 / 0.40 | 28 / 0.24 | 55 / 0.48 | 54 / 0.47 | -| -| -|
> |PRS-SR|0.64|0.78|0.67|0.58|0.78|0.65|0.48|0.80|0.68|
> |PRS-SPL|0.64|0.73|0.64|0.58|0.76|0.66|0.46|0.76|0.65|
>
> However, mitigation strategies that involve training or fine-tuning are substantially more time-consuming to extend to Uni-NaVid [J], which was open-sourced only a month ago and is still being integrated as an additional model for benchmarking. Reproducing its training pipeline is computationally prohibitive: the original paper estimates approximately 1,400 H800 GPU hours, which exceeds our available resources.

---

> ### Author Response · Authors · 2025-11-23
> **Response to Reviewer mYxG (Part 2)**
>
> ## Response to Question 1
>
> Thank you for pointing out this typo. We have updated Table 1 in the revised paper to show WMNav [E] indeed accepts the depth input modality.
>
> ## Response to Question 2
>
> Thank you for pointing out this discrepancy. We have updated the analysis in the **RGB Image Corruptions part in Section 4.2** in the revised paper: “This trend is observed with Black-out and Foreign Object corruptions. For the Black-out corruption, map-building agents (ETPNav and L3MVN) drop 10% and 3%, while RGB-only agents (NaVid and PSL) drop 37% and 30%, respectively. For the Foreign Object corruption, RGB-only agents (NaVid and PSL) drop 20% and 18%, respectively.”
>
> ## Response to Question 3
>
> Thank you for the helpful comment. We have revised the sentence in the **Instruction Corruptions part in Section 4.2** to: “Overall, the SR across different corruptions is consistent with the view that tokenization artifacts (e.g., masking, capitalization) and vocabulary coverage play a major role in robustness to instruction corruptions.” This revision removes the unsupported comparison and aligns the statement strictly with the evidence presented in Figure 4.
>
> ## References
>
> [A] Krantz, Jacob, et al. “Waypoint models for instruction-guided navigation in continuous environments." IEEE/CVF International Conference on Computer Vision (ICCV), 2021.
>
> [B] Ye, Joel, et al. “Auxiliary tasks and exploration enable objectnav.” arXiv preprint arXiv:2104.04112, 2021.
>
> [C] He, Diqi, et al. “STRIDER: Navigation via Instruction-Aligned Structural Decision Space Optimization.” Neural Information Processing Systems (NeurIPS), 2025
>
> [D] An, Dong, et al. “Etpnav: Evolving topological planning for vision-language navigation in continuous environments.” IEEE Transactions on Pattern Analysis and Machine Intelligence, 2024.
>
> [E] Nie, Dujun, et al. “Wmnav: Integrating vision-language models into world models for object goal navigation.” IEEE/RSJ International Conference on Intelligent Robots and Systems (IROS), 2025.
>
> [F] Yu, Bangguo, Hamidreza Kasaei, and Ming Cao. “L3mvn: Leveraging large language models for visual target navigation.” IEEE/RSJ International Conference on Intelligent Robots and Systems (IROS), 2023.
>
> [G] Yokoyama, Naoki, et al. “Vlfm: Vision-language frontier maps for zero-shot semantic navigation.” IEEE International Conference on Robotics and Automation (ICRA), 2024.
>
> [H] Zhang, Jiazhao, et al. “NaVid: Video-Based VLM Plans the Next Step for Vision-and-Language Navigation.” Robotics: Science and Systems, 2024.
>
> [I] Sun, Xinyu, et al. “Prioritized semantic learning for zero-shot instance navigation.” European Conference on Computer Vision, 2024.
>
> [J] Zhang, Jiazhao, et al. “Uni-NaVid: A Video-Based Vision-Language-Action Model for Unifying Embodied Navigation Tasks.” Robotics: Science and Systems, 2025.

---

### Author Response · Authors · 2025-12-03
**Summary for the New Area Chair**

**To the New Area Chair,**

We thank you for your time and effort in handling this submission. This paper introduces **NavTrust**, a benchmark evaluating the trustworthiness of embodied navigation agents under corrupted input settings. Below is a summary of the strengths, our improvements, and the review context prior to the score reversion.

### **1. Summary of Strengths**
We are encouraged that the reviewers came up with the strengths following:
1. **Novelty (mentioned by reviewer s9Yy and zNhG):** Fills a critical gap by jointly assessing perceptual and linguistic robustness under a unified framework, surpassing partial benchmarks like RobustNav.
2. **Contribution:**
- **a) Comprehensive Corruption Suite (mentioned by reviewer mYxG, s9Yy, GQf3, zNhG and u1M3)**: Introduces diverse, realistic distortions across RGB, depth, and instructions, covering noise, adversarial prompts, and stylistic rephrasing.
- **b) Mitigations for Corruptions (mentioned by reviewer mYxG, s9Yy, GQf3, zNhG and u1M3):** Goes beyond diagnosis by proposing concrete mitigation strategies for corruptions, offering both a testing framework and practical solutions.
- **c) Empirical Breadth (mentioned by reviewer s9Yy and u1M3):** Evaluates six state-of-the-art navigation agents (e.g., ETPNav, NaVid, VLFM), providing valuable comparative insights into model robustness.
**3. Reproducibility (mentioned by reviewer GQf3):** The work integrates corruptions into a unified evaluation platform, which encourages reproducibility of the work and community adoption.
4. **Clarity (mentioned by reviewer s9Yy, GQf3 and u1M3):** The paper is logically structured, easy to follow, and presents research results clearly.

### **2. Rebuttal & Improvements**
We have addressed all concerns with significant updates:
1. **Generality (mentioned by reviewer s9Yy and u1M3):** We incorporated the **RxR (Room-Across-Room)** dataset to assess robustness under multilingual and fine-grained instructions, expanding beyond R2R.
2. **Overhead (mentioned by reviewer s9Yy):** We added **Section 4.3** reporting the mean episode time for NavTrust versus clean evaluation to ensure transparency regarding computational costs.
3. **Validity (mentioned by reviewer GQf3):** We validated instruction attacks via **Gemini 2.5 Pro** and **human inspection**, confirming 95/100 semantic preservation of navigation goals.
4. **Baselines (mentioned by reviewer zNhG):** We added a new baseline: **Uni-NaVid**, a state-of-the-art VLA model, to compare map-implicit robustness against map-explicit modular baselines.
5. **Real-World Experiment (mentioned by reviewer GQf3 and zNhG):** We deployed **Uni-NaVid and ETPNav on a real robot** under both clean and corrupted settings. We verified our mitigation strategies indeed generalize to the real world. Demonstration videos are now included in the supplementary material.

### **3. Context on Score Reversion & Status**
We fully respect the ICLR Program Committee’s decision to revert score changes to preserve scientific integrity. However, we wish to share context regarding the upward trajectory of our reviews prior to the interruption:

* **Reviewer GQf3 (Original: 2):** On **November 25**, well before the incident was reported, this reviewer raised their score to **4**, noting that concerns were addressed but was waiting on real-world experiments. **We have resolved this** via the real-robot deployment mentioned above (Section 2, Point 5), though we did not have the opportunity to receive the follow-up response from the reviewer.
* **Reviewer zNhG (Score: 4):** Noted that most concerns were addressed but flagged "real-world deployment" as unresolved. **We have resolved this** via the real-robot deployment mentioned above (Section 2, Point 5), though we did not have the opportunity to receive the follow-up response from the reviewer.
* **Reviewer u1M3 (Score: 6):** Maintained the positive score, indicating all concerns were resolved.
* **Reviewers mYxG & s9Yy (Score: 6):** Both gave positive initial scores; we did not receive a response from them.

The discussion was moving in a positive direction, with the remaining critique (real-world validation) explicitly addressed in our final update. We appreciate your willingness to review these materials with fresh eyes and trust your judgment in making a fair recommendation.

Thank you for your dedication to the ICLR community.

---

### Meta-Review · Area_Chair_yrRj · 2025-12-11

**Summary:**

Scope/generalizability narrow: benchmark relies on Matterport3D and English; some reviewers want extra datasets (e.g., HM3D, RxR) and real-world validation [s9Yy, u1M3, zNhG]

Baseline coverage is thin: include more recent LLM/VLM or training-free agents (beyond the few VLN/OGN models used) [zNhG, mYxG]

Depth corruption evaluation: focus on map-building uses of depth; reviewers request to broaden methods; also, mitigations were analyzed mostly on ETPNav [mYxG]

Novelty/analysis: contributions feel incremental; corruptions look empirical; want clearer sensor/physics grounding or theory, and metrics beyond SR (PRS only) [GQf3, s9Yy]

Instruction corruptions: LLM-generated variants may not preserve semantics; request human/independent verification [GQf3]

Presentation/clarity nits: figures suggest wrong modality scope; odd depth visualization; PRS heatmap misses “clean” context; SR vs SPL notation; dataset/difficulty stats; WMNav depth input mismatch; wording around tokenization vs spatial-reasoning; NaVid blur inconsistency [zNhG, mYxG]

Compute/overhead not reported: need latency/throughput impact of corruptions/mitigations [s9Yy]

Why is VLFM so robust?: one reviewer mainly asks for architectural explanation [zNhG]

**Reviewer Concerns:**

Broader scope & generalization: Addressed. Added RxR (multilingual VLN) and HM3D (OGN) results; trends analyzed across datasets. [s9Yy, u1M3]

Real-world validation: Addressed. Deployed ETPNav and Uni-NaVid on a physical robot under clean/corrupted/mitigated settings; videos in supplement [zNhG, GQf3]

Baselines: Partially Addressed. Added Uni-NaVid (video-based VLA) and reported instruction-mitigation across multiple models, but training-heavy mitigations still limited by resources [zNhG, mYxG]

Depth corruption & “map-building” focus: Partially Addressed. Clarified terminology, motivated depth usage patterns, noted open-source constraints; added depth-adapter mitigation analysis, but no added non–map-building depth baselines in the benchmark [mYxG]

Novelty/analysis depth: Addressed. Provided sensor-physics grounding (e.g., CMOS noise, ToF multipath), motivation for observation-level corruptions, mechanistic analyses (memory vs reactive, fusion timing, VLM vs non-VLM), multi-intensity stress tests, and added PRS-SPL plus mCE/robustness-drop correlation [GQf3, s9Yy, u1M3]

Instruction-variant validity: Addressed. Verified with Gemini 2.5 Pro (95/100 semantic preservation) and human inspection [GQf3]

Compute/overhead: Addressed. New §4.3 with per-model episode times showing corruption-induced latency (e.g., low-light+noise spikes) [s9Yy]

VLFM robustness: Addressed. Gave architectural rationale (depth-driven mapping + BLIP-2 semantic scoring decoupled from RGB degradations) [zNhG]

**Reviewer Scores:**

As far as could be seen from the discussions, the final - visible - scores were 6, 6, 4 (from 2), 4 (stayed there 'most critical issue remains unresolved'), 6. Overall, there is no convincing support for this paper...

---

### Decision · Program_Chairs · 2026-01-26

Reject